# Chloroplast Hibernation-Promoting Factor PSRP1 Prevents Ribosome Degradation Under Darkness Independently of 100S Dimer Formation

**DOI:** 10.3390/plants14203155

**Published:** 2025-10-13

**Authors:** Kenta Tanaka, Yusuke Yoshizawa, Takashi Oda, Yasuhiko Sekine

**Affiliations:** 1Department of Life Science, Graduate School of Science, Rikkyo University, Nishi-Ikebukuro, Toshima-ku 171-8501, Tokyo, Japan; 2Materials and Life Science Division, J-PARC Center, JAEA, Tokai 319-1195, Ibaraki, Japan

**Keywords:** chloroplast, ribosome hibernation, ribosomal RNA, plastid-specific ribosomal protein, protein synthesis

## Abstract

Ribosome hibernation is a conserved translational stress response in bacteria, regulated by the hibernation-promoting factor (HPF). Plastid-specific ribosomal protein 1 (PSRP1) is the chloroplast ortholog of bacterial HPF. Although bacterial HPFs have been extensively characterized, both structurally and mechanistically, the physiological roles and mechanisms of PSRP1 in plant chloroplasts remain unclear. Here, we aimed to clarify the role of PSRP1 in chloroplast ribosome hibernation by examining its function under dark-stress conditions in the moss *Physcomitrium patens*. The PSRP1 knockout mutant exhibited moderate but statistically significant growth defects under both long- and short-day conditions compared to those of the wild-type plants. Moreover, the mutant displayed pronounced growth delay when co-cultured with wild-type plants, indicating a competitive disadvantage. Under dark conditions, wild-type plants exhibit increased PSRP1 protein accumulation, whereas the knockout mutant displayed reduction in chloroplast rRNA content. Notably, although PSRP1 is capable of inducing 100S dimers, we detected no chloroplast 100S dimers either in vivo or in vitro, suggesting a chloroplast-specific ribosome protection mechanism distinct from that of bacteria. These findings reveal PSRP1-mediated chloroplast ribosome protection and could provide new insights into plant stress tolerance.

## 1. Introduction

Chloroplasts are primary sources of fixed carbon and chemical energy in most ecosystems on Earth [1]. The endosymbiotic theory states that chloroplasts evolved when ancestral eukaryotic cells engulfed a cyanobacterium [2,3,4,5]. This primary event occurred approximately one billion years ago and led to the emergence of three evolutionary lineages: glaucophytes, rhodophytes (red algae), and chlorophytes (green algae). Land plants diverged from chlorophytes approximately 400–475 million years ago [6,7]. Throughout evolutionary history, numerous cyanobacteria-derived genes have migrated to the nuclear genome. However, some genes are retained within the chloroplast genome, where bacterial transcription and translation systems continue [4,8].

Protein synthesis, a critical process that accounts for nearly half of the cellular resources [9,10,11,12], is executed in chloroplasts by 70S ribosomes composed of 50S and 30S subunits, which retain characteristics inherited from the cyanobacterial ancestor. Under limited light and nutrient conditions, reducing protein synthesis in chloroplasts may be crucial for conserving cellular energy and nutrient reserves, ultimately promoting plant survival.

Under specific stress conditions, bacterial protein synthesis is regulated by various mechanisms, including new ribosome synthesis inhibition [13], excess ribosome degradation [14,15], and rapid modulation or suppression of translation by specific factors [16]. A striking change in the bacterial translation apparatus under stress, such as nutrient starvation, is 70S ribosome dimerization to form 100S complexes [17,18]. This mechanism, called ribosome hibernation [19], occurs in almost all bacteria and is classified into two major types [20,21]. In *γ*-proteobacteria, including *Escherichia coli*, ribosome hibernation occurs when ribosome modulation factor (RMF) binds to 70S ribosomes. This process interferes with the Shine–Dalgarno sequence and induces conformational changes in the 30S subunit, forming a 90S ribosomal dimer. Short-form hibernation-promoting factor (SHPF) then binds to the decoding center, where mRNA and tRNA interact, facilitating conversion of the 90S dimer intermediate to the 100S dimer [20,22]. In contrast, many other bacteria lack RMF and possess long-form HPF (LHPF), a SHPF homolog with an additional C-terminal domain (CTD) that promotes 100S dimer formation [17,23]. LHPF forms a homodimer via the CTD and triggers 100S dimer formation by tethering the two 30S subunits [24,25,26]. In addition, cyanobacteria, which are considered the evolutionary origin of chloroplasts, also harbor an ortholog of HPF known as LrtA. This gene was initially identified as a light-repressed transcript, since its mRNA accumulates in darkness but is rapidly degraded upon illumination [27]. A phylogenetic analysis revealed that LrtA is more closely related to LHPF than to SHPF [28]. Consistent with this relationship, LrtA retains the ability to promote the formation of 100S ribosome dimers, thereby supporting a conserved role in ribosome stabilization [21,29].

In 1990, a putative HPF encoded in the nuclear genome that binds to the 30S subunit of the chloroplast ribosome was identified [30]. Since bacterial HPFs were not reported until the 2000s [22,31], this protein was initially thought to be chloroplast-specific and was thus named plastid-specific ribosomal protein 1 (PSRP1). However, subsequent cryo-electron microscopy (cryo-EM) studies of spinach chloroplast ribosomes revealed that PSRP1 binds to the decoding center of chloroplast ribosomes, similar to SHPF and LHPF [32]. Furthermore, phylogenetic analysis has revealed that PSRP1 is an LHPF ortholog [20]. The high resolution cryo-EM microscopy structures of spinach chloroplast ribosome-bound PSRP1 has been determined [32,33,34,35,36]. Particularly, using PSRP1 with the cryo-EM spinach chloroplast ribosomes by Boerema et al. (2018) revealed that bS1c is reduced and tightly bound via an N-terminal extension, precluding HPF-like chloroplast ribosome dimerization by occupying the interface [36]. Despite the potential structural importance of PSRP1, its physiological functions under specific stress conditions and precise biological function have yet to be fully elucidated.

In the present study, we comprehensively analyzed PSRP1 function using the model organism *Physcomitrium patens*, while emphasizing its role under dark-stress conditions. As a basal land plant, *P. patens* occupies an evolutionary position bridging cyanobacteria and higher plants, providing a unique opportunity to link insights from cyanobacterial LrtA to the function of PSRP1 in plants.

## 2. Results

### 2.1. PSRP1 Is a Well-Conserved Chloroplast Protein Throughout Plant Evolution

Using the amino acid sequence of spinach PSRP1, we searched plant genomes and confirmed the presence of PSRP1, from green algae to seed plants (Appendix A). Furthermore, our phylogenetic analysis of PSRP1 shows that its evolutionary lineage aligned with major lineages of plants, from green algae to seed plants (Appendix A). This demonstrates that plant PSRP1 genes are closely related to those of cyanobacteria, such as *Synechococcus* and *Synechocystis*, the ancestors of chloroplasts. This relationship supports the hypothesis that PSRP1 was inherited from cyanobacterial ancestors through endosymbiosis and conserved throughout plant evolution. The extensive evolutionary conservation of this gene suggests that PSRP1 is crucial for plant survival in competitive environments. To gain further insights into its function, we analyzed PSRP1 using the bryophyte model organism *P. patens*.

NCBI database analysis revealed two *PSRP1* genes in *P. patens*, which we designated *PSRP1a* (NCBI Gene ID: 112274408) and *PSRP1b* (NCBI Gene ID: 112295268). *PSRP1a* and *PSRP1b* amino acid sequences share 94% identity, excluding the N-terminal chloroplast transit peptide (Appendix A). We confirmed *PSRP1a* and *PSRP1b* mRNA and the presence of PSRP1 protein in *P. patens* using reverse transcription PCR (RT-PCR) and Western blot (Figure 1A,B). To investigate intracellular PSRP1 localization, we constructed strains expressing PSRP1a-GFP or PSRP1b-YFP fusion proteins controlled by their native promoters. Fluorescence microscopy showed that both PSRP1a-GFP and PSRP1b-YFP were localized in chloroplasts (Figure 1C,D). These results suggest that the *P. patens* genome harbors two *PSRP1* gene copies, which encode functional proteins localized in chloroplasts.

### 2.2. Loss of PSRP1 Impairs Plant Growth

To investigate the physiological role of PSRP1 in plants, a double-knockout strain was generated by replacing *PSRP1a* and *PSRP1b* with *nptII* and *aphIV*, respectively. We confirmed the absence of the mRNAs and protein expression (Figure 1A,B). This double-knockout strain is referred to as *PSRP1a/b*. We compared the growth of the wild type and *PSRP1a/b* strains by cultivating protonemata on agar plates under long-day conditions (LD; 16 h light/8 h dark) and short-day conditions (SD; 8 h light/16 h dark) conditions for over 6 weeks. No differences in growth were observed during the early growth stages. However, after day 46 under LD conditions and day 36 under SD conditions, the *PSRP1a/b* strain showed statistically significant albeit modest decreases in growth relative to that of the wild type (Figure 2A,B, *p* < 0.05) under both conditions. To simulate natural environmental conditions, such as limited light, nutrients, and space, we co-cultured wild-type and *PSRP1a/b* protonemata under LD and SD conditions. After 3 weeks, real-time PCR of the extracted plant genomic DNA amplified strain-specific sequences: the *PSRP1a* gene for the wild type and the *nptII* gene for *PSRP1a/b*. The relative amplification levels of *PSRP1a* and *nptII* were normalized to the internal control (*Actin*) to estimate *PSRP1a/b* abundance relative to the wild type. After 3 weeks, the survival ratio of *PSRP1a/b* relative to that of the wild type decreased significantly under both conditions, indicating that PSRP1 plays a crucial role in plant growth, particularly in competitive environments.

### 2.3. PSRP1 Accumulates and Plays a Protective Role in Preventing Degradation of Chloroplast Ribosomes Under Dark Conditions

Bacterial HPFs associate with ribosomes under specific stress conditions or during stationary phases [18,21,37,38,39,40,41,42,43]. As shown in Appendix A, PSRP1 is an HPF ortholog that binds to chloroplast ribosomes under certain stress conditions, potentially modulating their activity. However, as shown in Figure 2, despite the absence of specific stress, the growth rates differed significantly. Thus, we hypothesized that darkness may act as a stressor that triggers the function of PSRP1. To investigate the effect of dark conditions on plant growth and the dynamics of PSRP1 protein, plants were exposed to prolonged darkness. To evaluate plant growth, wild-type and *PSRP1a/b* knockout lines were co-cultivated under dark conditions for 2 d within a one-week growth cycle, which was repeated for three consecutive weeks (Appendix A). Despite the severity of these conditions, growth differences were still observed in the wild-type and *PSRP1a/b* lines under competitive conditions (Appendix A).

Next, we examined the changes in PSRP1 protein levels under dark conditions. Plants were grown for a single week in darkness, lasting for 1, 2, or 3 d before sample collection (Figure 3A). The analysis revealed significant protein accumulation with increasing duration of dark exposure (Figure 3B).

Total RNA was extracted from plants grown under the conditions shown in Figure 3A, and RT-qPCR was performed to assess chloroplast rRNA levels. In wild-type plants, the chloroplast (Cp) 16S and 23S rRNA amounts did not differ significantly, regardless of the duration of darkness. In contrast, the *PSRP1a/b* strains exhibited significantly decreased Cp16S and Cp23S rRNA levels with increasing duration of darkness (Figure 3C,D). Consistent with these results, RNA gel blot analysis demonstrated degradation of chloroplast rRNAs (Cp16S and Cp23S), while immunoblot analysis further revealed that the levels of chloroplast ribosomal protein S1 (bS1c) were also decreased in the *PSRP1a/b* strains under prolonged darkness (Figure 3E,F). These results suggest that PSRP1 prevents chloroplast rRNA degradation in the dark.

### 2.4. Absence of 100S Dimers in Chloroplast Ribosomes Despite PSRP1 Dimerization and Induction of 100S Formation in Bacterial Ribosomes

In bacteria, HPF binding facilitates 100S dimer formation, which is resistant to RNase R-induced degradation [44,45]. Although our alignment results indicate that PSRP1 is closely related to LHPF (Appendix A) and it is associated with chloroplast ribosomes in *P. patens*, no 100S dimers were detected under either 0-d or 2-d dark conditions (Appendix A). LHPF dimerization is critical for 100S ribosomal dimer formation [24,46]. Therefore, we examined whether PSRP1 dimerizes.

To determine the oligomeric states of PSRP1a and PSRP1b, we performed SEC-MALS, which is a reliable method for measuring absolute molar mass in solution. After removing the presumed signal peptides, mature PSRP1a and PSRP1b were found to comprise 219 and 221 residues, respectively. SEC-MALS revealed molecular masses of 48.2 kDa and 53.6 kDa, roughly twice their predicted masses of 26.7 and 26.8 kDa (Figure 4A,B). As a control, *Bacillus subtilis* LHPF (*Bs*HPF), which forms dimers, was analyzed and showed a dimeric mass of 44.3 kDa, approximately twice the predicted value of 24.2 kDa (Figure 4C). Additionally, BSA had a mass of 61.2 kDa, which closely matches the predicted value (66.4 kDa) (Figure 4D). This analysis demonstrates the reliability of SEC-MALS and confirms that PSRP1 forms a stable dimer in solution. The SEC-MALS results, including predicted and observed masses and oligomeric states, are summarized in Table 1.

Next, we tested whether PSRP1 dimers induce 100S ribosome dimerization in vitro. Chloroplast ribosomes purified from spinach were mixed with purified PSRP1a protein from *P. patens* at ratios of 1:1 and 1:5 in vitro. Although both endogenously bound PSRP1 from spinach and purified *P. patens* PSRP1a bound to chloroplast ribosomes, no chloroplast 100S dimer formation was observed (Figure 5A–C). This is consistent with the spinach chloroplast ribosome structure, in which bS1c occupies the anticipated dimer interface, thereby blocking 100S formation [36]. In contrast, incubation of PSRP1a with ribosomes purified from a *Lactococcus lactis hpf* knockout strain (*Δlhpf*) in vitro resulted in 100S dimer formation (Figure 5D–F). Thus, although PSRP1 retains intrinsic 100S-promoting capacity on bacterial ribosomes, chloroplastic bS1c likely prevents dimerization, which agrees with the prior structural model.

## 3. Discussion

We investigated the physiological role of chloroplast PSRP1 by focusing on its response to dark stress. Our results show that *PSRP1*-deficient plants (*PSRP1a/b*) exhibited growth defects after more than one month under both LD and SD conditions when grown individually. In contrast, when co-cultivated with wild-type plants, the *PSRP1a/b* strain became more evident over a much shorter period. Furthermore, PSRP1 levels increased under dark conditions, and contributed to maintaining chloroplast ribosome content. This mechanism may enable reuse of inactive ribosomes when the stress conditions improve. Consequently, PSRP1 contributes to faster regrowth upon return of light, which likely accounts for the growth differences between wild-type and *PSRP1a/b* plants. Thus, evolutionary conservation of PSRP1 may be crucial for plant survival under harsh environmental conditions. A previous study on maize, however, observed plant growth over 7–9 d under a 16-h light/8-h dark cycle and reported that PSRP1 absence did not affect growth. No significant changes were found in PSRP1 protein abundance under light and dark conditions [47]. These discrepancies could be explained not only by differences in plant species (C4 maize versus C3 bryophyte *P. patens*), but also by the severity and duration of dark stress applied. While Swift et al. (2020) [47] monitored plants for only 7–9 d under standard LD cycles, our study involved not only prolonged dark treatments lasting several days but also repeated cycles of darkness extending over several weeks.

SHPF and RMF induce 100S dimer formation in *E. coli* and other *γ-proteobacteria* [22,48,49]. Most other bacteria possess LHPF, which retains an additional CTD. LHPF forms a homodimer via its CTD and induces 100S ribosome dimer formation by tethering the two 30S subunits [24,25,26,50,51]. These 100S dimers are more resistant to degradation than 70S ribosomes [44,45,52]. In contrast, our results show that chloroplast 100S dimer formation was not observed in vitro or in vivo (Figure 5A–C and Appendix A), although PSRP1 dimerized (Figure 4) and induced 100S dimer formation in bacterial ribosomes (Figure 5D–F). Boerema et al. found that the chloroplast ribosomal protein bS1c was tightly bound to the ribosome [36]. This cryo-EM analysis further revealed that bS1c is reduced and tightly bound via an N-terminal extension, with its binding region overlapped with the 30S–30S interface of the 100S dimer, thereby potentially inhibiting dimer formation (Appendix A). *P. patens* harbors the *bS1c* gene, and its sequence is well conserved with those of spinach and *A. thaliana* (Appendix A). Structural comparison of *P. patens* and spinach bS1c based on AlphaFold3 predictions confirmed that the overall fold is highly conserved (Appendix A). Therefore, inhibiting 100S dimer formation by bS1c is likely applicable to *P. patens*, consistent with our findings. These observations suggest that chloroplasts have evolved a ribosome protection mechanism distinct from that of bacteria.

At the same time, our study has some limitations. First, although we observed a clear reduction in chloroplast rRNA levels in *PSRP1a/b* strains, we cannot fully exclude alternative explanations such as reduced transcription or defects in rRNA processing. Second, our analyses were conducted primarily under prolonged darkness, and it remains to be determined whether PSRP1 has similar functions under other physiological stress conditions. Finally, this study focused on *P. patens*, and further investigations in other plant species will be important to determine whether the role of PSRP1 in maintaining chloroplast ribosome integrity is conserved across the plant kingdom.

An important direction for future research will be to elucidate the molecular basis underlying the absence of 100S dimer formation—potentially involving factors such as bS1c—and to clarify how PSRP1 protects ribosomes despite the lack of 100S dimers, as well as to determine whether these mechanisms operate under other environmental stresses and across diverse plant lineages.

In conclusion, our analysis revealed that PSRP1 stabilizes chloroplast ribosomes under dark conditions in the absence of 100S ribosome dimer formation, in contrast to HPF-mediated ribosome hibernation in bacteria. Our findings advance the understanding of chloroplast ribosome regulation and highlight the evolutionary divergence of plastid translation from its bacterial ancestors.

## 4. Materials and Methods

### 4.1. Plant Materials and Growth Conditions

*Physcomitrium patens* (Hedw.) Bruch & Schimp. was used as the model organism in this study. Protonemata were cultivated on BCDAT agar medium [53] at 25 °C.

### 4.2. Construction of a P. patens PSRP1a and PSRP1b Double Knockout Strain

To create the *PSRP1a* and *PSRP1b* double knockout strain, the 5′ and 3′ flanking regions of each gene were amplified by PCR using the primers listed in Appendix A. For PSRP1a, the 5′ region was amplified using PSRP1a-5′HR-KpnI Fwd and PSRP1a-5′HR-XhoI Rev, and the 3′ region with PSRP1a-3′HR-SacII Fwd and PSRP1a-3′HR-SacI Rev; these fragments were cloned into pTN3 [53] to construct pHAC7. For PSRP1b, 5′ region was amplified with PSRP1b-5′HR KpnI Fwd and PSRP1b-5′HR-XhoI Rev, and 3′ region with PSRP1b-3′HR SmaI Fwd and PSRP1b-3′HR-SacI Rev; the resulting fragments were inserted into pTN186 (a gift from Mitsuyasu Hasebe, Chiba, Japan, Addgene plasmid #34890) to yield pHAC8. The resulting constructs were linearized and introduced into *P. patens* protoplasts via PEG-mediated transformation [53]. Transformants were selected on BCDAT agar plates containing antibiotics and the insertion of a single copy of each gene was confirmed by PCR. The double knockout strain is referred to as *PSRP1a/b*.

### 4.3. Reverse Transcription-Polymerase Chain Reaction (RT-PCR)

Total RNA was extracted using an RNeasy^®^ Plant Mini Kit (Qiagen, Hilden, Germany) following the manufacturer’s protocol. Reverse transcription was conducted using ReverTra Ace^®^ qPCR RT Master Mix (Toyobo, Osaka, Japan) to generate cDNA from the isolated RNA. Gene-specific primers targeting *PSRP1a*, *PSRP1b*, and internal control Actin were used for subsequent PCR amplification. The PCR products were separated by agarose gel electrophoresis and visualized under UV light to verify the presence and expected size of the amplicons. The primer sequences are listed in Appendix A.

### 4.4. Fluorescence Microscopy

The *PSRP1a-sGFP* knock-in plasmid (pGFPmutNPTII-PSRP1a) was constructed by amplifying the 5′ and 3′ homologous regions (5′HR and 3′HR) from the *P. patens* genome using primer pairs PSRP1a-5′HR Fwd/Rev and PSRP1a-3′HR Fwd/Rev. The resulting PCR fragments were digested with SalI/ClaI (5′HR) and XbaI/SacI (3′HR). These fragments were then ligated into the corresponding sites of the pGFPmutNPTII vector [54]. pCTRN-aphIV was constructed by replacing the antibiotic resistance gene in pCTRN-NPTII 2 (NCBI Accession No. AB697058) with the *aphIV* gene. Specifically, *aphIV* was amplified from pTN186 using aphIV Fwd/Rev primers, while pCTRN-NPTII 2 was amplified with pCTRN-NPTII 2 Fwd/Rev primers. The resulting fragments were assembled using a NEBuilder^®^ HiFi DNA Assembly Kit. Using the *P. patens* genome as the template, 5′HR and 3′HR were amplified with PSRP1b-5′HR Fwd/Rev and PSRP1b-3′HR Fwd/Rev, respectively. In parallel, fragments No.1 and No.2 of pCTRN-aphIV were amplified using pCTRN-aphIV No.1 Fwd/Rev and pCTRN-aphIV No.2 Fwd/Rev. These fragments were subsequently assembled with NEBuilder^®^ HiFi DNA Assembly Kit to generate the final plasmid, pTaK16. All primer sequences are listed in Appendix A. The plasmids were linearized and introduced into plant cells via polyethylene glycol (PEG)-mediated protoplast transformation [53]. Transformants were selected on BCDAT agar plates containing antibiotics, and the insertion of a single copy of each gene was confirmed by PCR. The intracellular localization of PSRP1a-GFP or PSRP1b-Citrine in *P. patens* was observed using an LSM 800 microscope (Carl Zeiss, Jena, Germany). Additionally, chlorophyll autofluorescence was monitored to assess its localization within the chloroplasts.

### 4.5. Western Blotting

The protonemata were collected, drained, and immediately frozen in liquid nitrogen. Tissues were homogenized at 1750 rpm for 10 s using a Multi-beads shocker^®^ (Yasui Kikai, Osaka, Japan). Total protein was extracted using extraction buffer (200 mM Tris-HCl, pH 8.0, 200 mM KCl, 35 mM MgCl_2_, 25 mM EGTA, 1% Triton X, 100 mM β-mercaptoethanol (β-ME), 2 mM PMSF). Cell debris was removed by centrifugation at 17,800× *g* for 20 min at 4 °C. The supernatant was analyzed by sodium dodecyl-sulfate polyacrylamide gel electrophoresis (SDS-PAGE) on a 15% polyacrylamide gel, followed by immunoblotting with polyclonal anti-PSRP1 antibody (1/1000, custom-made) and polyclonal anti-bS1c antibody (1/1000, PhytoAB, San Jose, CA, USA). Alkaline phosphatase-conjugated goat anti-rabbit IgG (1/10,000, Sigma-Aldrich, St. Louis, MO, USA) was used as the secondary antibody, and detection was performed with AttoPhos^®^ substrate (Promega, Madison, WI, USA) using a Typhoon™ FLA 9500 imager (GE Healthcare, Chicago, IL, USA). Band intensity of PSRP1 was measured from scanned immunoblot images using ImageJ software (Fiji, version 2.9.0; Image J 1.54p, https://imagej.net/ij/), accessed on 29 July 2020.

### 4.6. Northern Blotting

Total RNA was electrophoresed on MOPS-formaldehyde denaturing 1.2% agarose gels and transferred onto a Hybond™ N^+^ membrane (GE Healthcare, Chicago, IL, USA) via capillary transfer for at least 3 h. RNA was detected using a DIG-labeled probe with DIG reagents and kits for nonradioactive nucleic acid labeling and detection (Roche, Basel, Switzerland), according to the manufacturer’s protocol. The probe sequences are listed in Appendix A.

### 4.7. Growth Comparison and Competitive Survival Analysis

For growth comparisons, small pieces of protonemal cells were inoculated on BCDAT agar plates and cultivated at 25 °C under long-day (LD; 16 h light/8 h dark) and short-day (SD; 8 h light/16 h dark) conditions for over 1 month. For competitive survival analysis, the wild-type and *PSRP1a/b* strains were grown separately under continuous white light for 7 d. Wild-type and *PSRP1a/b* strains were combined at equal fresh weights, homogenized, and spread on BCDAT agar plates. The cultures were incubated under LD or SD conditions for three weeks. For growth comparisons under prolonged dark conditions, the cultures were first grown under continuous light for 5 d, followed by 2 d in the dark, homogenized again, and spread on fresh BCDAT agar plates. This process was repeated weekly for 3 weeks. Genomic DNA was extracted using cetyl-trimethyl-ammonium bromide [55]. Real-time PCR was performed using PowerUp™ SYBR™ Green Master Mix (Thermo Fisher Scientific, Waltham, MA, USA) and a QuantStudio™ 12 K Flex Real-Time PCR System (Thermo Fisher Scientific) to amplify strain-specific sequences (*PSRP1a* for wild-type and *nptII* for *PSRP1a/b*), with *Actin* serving as the internal control. Sequences of the gene-specific primers are listed in Appendix A.

### 4.8. Real-Time Quantitative PCR (RT-qPCR)

Total RNA was extracted using the RNeasy^®^ Plant Mini Kit (Qiagen, Hilden, Germany) according to the manufacturer’s instructions. Complementary DNA (cDNA) was synthesized from the isolated RNA using ReverTra Ace^®^ qPCR RT Master Mix (Toyobo, Osaka, Japan). The cDNA was subsequently amplified using PowerUp™ SYBR™ Green Master Mix (Thermo Fisher Scientific) on a QuantStudio™ 12 K Flex Real-Time PCR System (Thermo Fisher Scientific), following the manufacturer’s guidelines. The abundance of chloroplast 16*S* and 23*S rRNA* were normalized to mitochondrial *18S rRNA*. The sequences of gene-specific primers are listed in Appendix A.

### 4.9. Protein Purification

*PSRP1a* and *PSRP1b* from *P. patens* were amplified from the cDNA, whereas *BsHPF* from *B. subtilis*, *LlHPF* from *L. lactis,* and *EcHPF* from *E. coli* were amplified from the genomic DNA. Each gene was cloned into pET15b vector using gene-specific primers and the appropriate restriction enzyme sites. Details of the primers used for each gene are summarized in Appendix A. The recombinant plasmid, pET15b, was transformed into *E. coli* strains ER2566 and BL21(DE3) for protein expression. Cultures were grown in LB medium supplemented with 0.2% glucose and 100 µg/mL ampicillin at 37 °C until the optical density at 600 nm (OD_600_) reached 0.4–0.5, at which point protein expression was induced by adding 0.4 mM isopropyl β-D-1-thiogalactopyranoside. Following induction, the cultures were incubated for another 3 h at 37 °C before harvesting the cells via centrifugation. The cells were resuspended in lysis buffer (50 mM Tris-HCl, pH 7.5; 500 mM NaCl; 10% (*w*/*v*) sucrose; 5 mM β-ME), frozen in liquid nitrogen, and stored at −80 °C. Lysis was performed by freeze-thawing with lysozyme (1 mg/mL). Cell debris was removed by centrifugation at 40,000× *g* for 30 min at 4 °C. Proteins were purified using TALON^®^ Metal Affinity Resin (Takara Bio, Shiga, Japan) and eluted in a buffer containing 20 mM Tris-HCl (pH 8.0), 500 mM NaCl, 5 mM β-ME, and 100 mM imidazole. Protein concentrations were measured at 280 nm using a NanoDrop™ 2000 spectrophotometer (Thermo Fisher Scientific).

### 4.10. Size-Exclusion Chromatography with Multi-Angle Light Scattering (SEC-MALS)

SEC-MALS measurements were performed using a DAWN^®^ HELEOS^®^ II (Wyatt Technology Corporation, Santa Barbara, CA, USA) downstream of an Alliance liquid chromatography system (Waters, Milford, MA, USA) connected to a Superdex™ 200 5/150 GL (Cytiva, Uppsala, Sweden) gel filtration column. The differential refractive index downstream of the MALS detector was used to determine protein concentration. The column was equilibrated with a running buffer containing 100 mM Tris-HCl (pH 8.0), 150 mM NaCl, 1 mM EDTA, and 10% (*v*/*v*) glycerol. The flow rate was set to 0.15 mL/min, and 90 µL of each sample at 1.6 mg/mL was injected. Data analysis was performed using ASTRA version 6 software (Wyatt Technology Corporation, Santa Barbara, CA, USA).

### 4.11. Isolation of Spinach Chloroplast 70S Ribosomes

Chloroplast ribosomes were isolated as described by Bartsch et al. [56]. Briefly, 4 kg of spinach leaves were purchased from a local supermarket. The leaves were homogenized using Buffer A (10 mM Tris-HCl, pH 7.5, 50 mM KCl, 10 mM Mg(OAc)_2_, 6 mM β-ME) containing 0.7 M sorbitol. The homogenate was filtered through four layers of gauze. The suspension was cleared by centrifugation at 1200× *g* for 15 min. The pellet was resuspended in Buffer A with 0.4 M sorbitol and re-centrifuged at 1200× *g* for 15 min. This step was repeated thrice. The washed chloroplast pellet was resuspended in Buffer A supplemented with 2% (*v*/*v*) Triton X-100 and incubated at 4 °C for 30 min. The lysed suspension was clarified by centrifugation at 26,000× *g* for 30 min at 4 °C. This step was repeated. The supernatant was layered onto a 30% sucrose cushion in Buffer II (20 mM Tris-HCl, pH 7.5, 10 mM Mg(OAc)_2_, 1 M NH_4_OAc, 6 mM β-ME) and centrifuged at 206,000× *g* for 3 h at 4 °C. The green pellet was resuspended in Buffer II. After mixing for 1 h at 4 °C, the high-salt washed ribosomes were layered onto a 30% sucrose cushion in Buffer I (20 mM Tris-HCl, pH 7.5, 15 mM Mg(OAc)_2_, 100 mM NH_4_OAc, 6 mM β-ME) and centrifuged at 206,000× *g* for 4 h at 4 °C. The pellet was resuspended in Buffer I.

### 4.12. Isolation of Lactococcus lactis Ribosomes

The *Lactococcus lactis* strain NZ9000 *ΔyfiA* used in this experiment was kindly provided by Prof. Bert Poolman (University of Groningen) and originally constructed in a previous study [39]. *L. lactis* 70S ribosomes were isolated as described by Ueta et al. [21]. *L. lactis Δlhpf* strains were cultured statically at 30 °C in M17 medium (Difco Laboratories, Detroit, MA, USA) supplemented with 0.5% (*w*/*v*) glucose. After reaching an OD_600_ of 1.5, the cells were harvested by centrifugation at 5210× *g* for 15 min at 4 °C. The resulting cell pellets were ground with approximately equal volumes of quartz sand (Wako Pure Chemical Industries) and extracted using Buffer I. The homogenate was subjected to centrifugation at 9000× *g* for 15 min at 4 °C, and this step was repeated to ensure thorough separation. The resulting supernatant was layered onto a 30% sucrose cushion prepared in Buffer II. Ribosome washing and subsequent purification steps were performed as outlined under “Section 4.11. Isolation of Spinach Chloroplast Ribosomes.”

### 4.13. Analysis of 100S Dimers of Chloroplasts and L. lactis Ribosomes

Reactions were performed as described by Usachev et al. [57]. For the in vitro reaction, 64 pmol of chloroplast ribosomes was mixed with 64 or 320 pmol of purified PSRP1a protein. Similarly, 64 pmol *L. lactis* ribosomes were mixed with 64 pmol of purified LlHPF or PSRP1a protein. After 30 min of incubation in Buffer I (20 mM Tris-HCl pH 7.5, 15 mM Mg(OAc)_2_, 100 mM NH_4_OAc, 6 mM β-ME) at 37 °C, the complexes were layered onto a 5–20% sucrose gradient prepared in Buffer I and centrifuged at 11,900 rpm for 20 h at 4 °C using a Hitachi P40ST rotor. The gradient was fractionated using a Piston Gradient Fractionator (BioComp, San Antonio, TX, USA), and the absorbance at 254 nm was monitored using a Bio-mini UV Monitor (ATTO, Amherst, NY, USA). Each fraction was subjected to Western blot.

## Figures and Tables

**Figure 1 plants-14-03155-f001:**
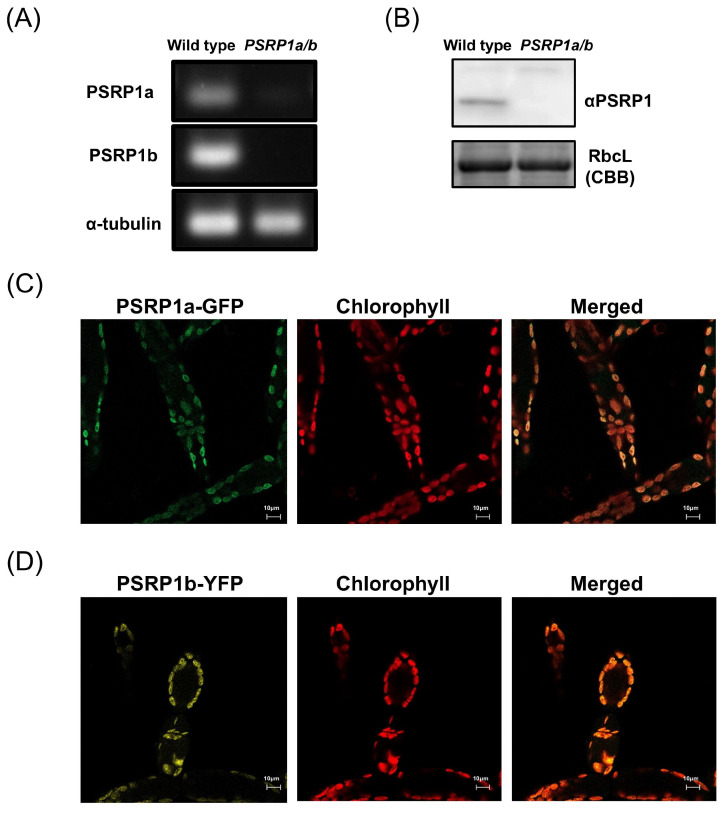
Expression and chloroplast localization of PSRP1 in *Physcomitrium patens*. (**A**) *PSRP1a* and *PSRP1b* mRNA expression analysis by RT-PCR in wild-type and *PSRP1a/b* strains. The amplification products of *PSRP1a*, *PSRP1b* mRNA and internal control (*α-tubulin* mRNA) were shown. No detectable *PSRP1a* and *PSRP1b* mRNA was observed in *PSRP1a/b* strain. (**B**) Detection of PSRP1 protein by Western blot in wild-type and *PSRP1a/b* strains. Detection was performed using a specific antibody against PSRP1 protein. RbcL protein stained with Coomassie Brilliant Blue (CBB) was also shown as an internal control. No detectable PSRP1 protein was observed in the *PSRP1a/b* strain. (**C**,**D**) Intracellular localization of PSRP1a-GFP (**C**) and PSRP1b-YFP (**D**) fusion proteins in plant cells. GFP (green) or YFP (yellow) fluorescence co-localized with chlorophyll autofluorescence (red), confirming that PSRP1a or PSRP1b fusion proteins were localized to the chloroplasts. In the merged images, overlapping signals appear orage. Bars = 10 μm in each image.

**Figure 2 plants-14-03155-f002:**
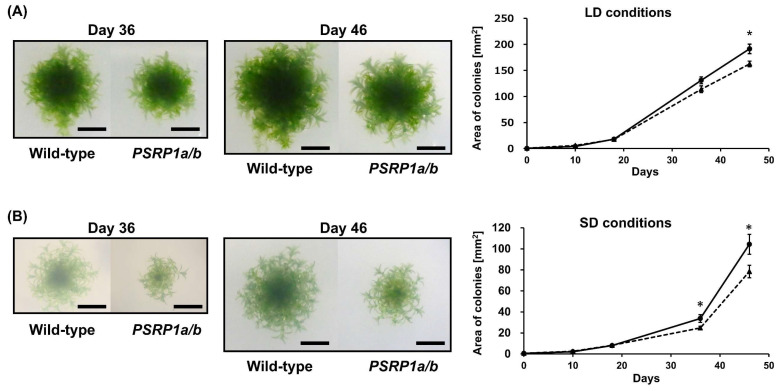
Comparative growth of wild-type and *PSRP1a/b* mutant lines in separate and competitive co-cultures. (**A**) Separate culture under long-day (LD) conditions: Representative colonies of wild-type and *PSRP1a/b* strains grown on BCDAT agar plates for 46 d (16 h light/8 h dark). Images taken on Days 36 and 46 are shown. Scale bar = 5 mm. The graph on the right shows the average colony area ± SE (n = 10). Solid line, wild type; dotted line, *PSRP1a/b*. A significant difference was observed on Day 46 (*p* < 0.05, Student’s *t*-test), as indicated by an asterisk (*). (**B**) Separate culture under short-day (SD) conditions: Representative colonies of wild-type and *PSRP1a/b* strains grown on BCDAT agar plates for 46 d (8 h light/16 h dark). Images taken on Days 36 and 46 are shown. Scale bar = 5 mm. The graph on the right shows the average colony area ± SE (n = 9). Solid line, wild-type; dotted line, *PSRP1a/b*. Significant differences were observed on Days 36 and 46 (*p* < 0.05, Student’s *t*-test), as indicated by an asterisk (*). (**C**) Schematic of the co-culture assay: Wild-type and *PSRP1a/b* strains were homogenized, mixed, and spread on BCDAT agar plates. The plates were then incubated for 3 weeks under either LD (16 h light/8 h dark) or SD (8 h light/16 h dark) conditions. Genomic DNA was extracted and used for real-time PCR analysis with strain-specific primers (*PSRP1a* for wild-type, *nptII* for *PSRP1a/b*), and *Actin* as an internal control. (**D**) Co-culture under LD conditions: The graph shows the average relative strain ratio (wild type: *PSRP1a/b*) at 0 and 3 weeks (n = 3). Solid line, wild-type; dotted line, *PSRP1a/b*. Asterisks (**) indicate a significant difference at 3 weeks (*p* < 0.05, Student’s *t*-test). (**E**) Co-culture under SD conditions: The graph shows the average relative strain ratio (wild-type: *PSRP1a/b*) at 0 and 3 weeks (n = 3). Solid line, wild-type; dotted line, *PSRP1a/b*. Asterisks (*) indicate a significant difference at 3 weeks (*p* < 0.05, Student’s *t*-test).

**Figure 3 plants-14-03155-f003:**
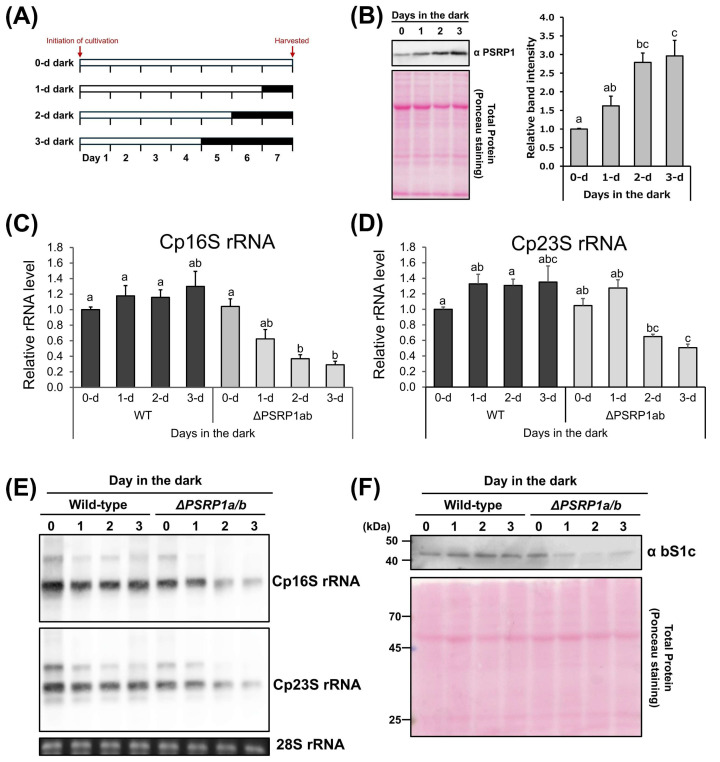
Effects of prolonged darkness on PSRP1 levels and Cp rRNA levels in wild-type and *PSRP1a/b* in *P. patens*. (**A**) Schematic representation of the growth conditions in this assay. The growing period for each condition was 7 d, with the time in darkness varying from 0 to 3 d. The white and dark gray bars represent continuous light and continuous darkness, respectively. (**B**) The left panel shows representative Western blot images. Detection was performed using a specific antibody against PSRP1. Total protein stained with Ponceau S is also shown as an internal control. The graph on the right shows the relative amount of PSRP1 protein under each condition. The data are presented as means + standard error (n = 3). Significant differences were identified by Tukey’s HSD test, with different letters indicating significance (*p* < 0.05). (**C**,**D**) Quantification of chloroplast rRNA in the wild-type and *PSRP1a/b* strains. Total RNA was extracted from the plants grown under each condition. Real- time quantitative PCR (RT-qPCR) was performed using primers specific to chloroplast 16S (**A**) or 23S rRNA (**B**), with mitochondrial 18S rRNA (mt18S rRNA) serving as the internal control. Amplification ratios relative to mt18S rRNA were used to estimate the amount of rRNA per cell. The graph shows the average ratio and standard error of the mean (n = 4). Significant differences were identified using Tukey’s HSD test, with different letters indicating statistical significance (*p* < 0.05). (**E**) RNA gel blot analysis of chloroplast 16S and 23S rRNAs in *P. patens*. Total RNA was separated on agarose gels and hybridized with probes specific to Cp16S and Cp23S rRNAs. Ethidium bromide–stained cytosolic 28S rRNA is shown as a loading control. (**F**) Western blotting of chloroplast ribosomal protein S1 (bS1c) in wild-type and *PSRP1a/b* strains. Ponceau S staining of total protein is shown as a loading control.

**Figure 4 plants-14-03155-f004:**
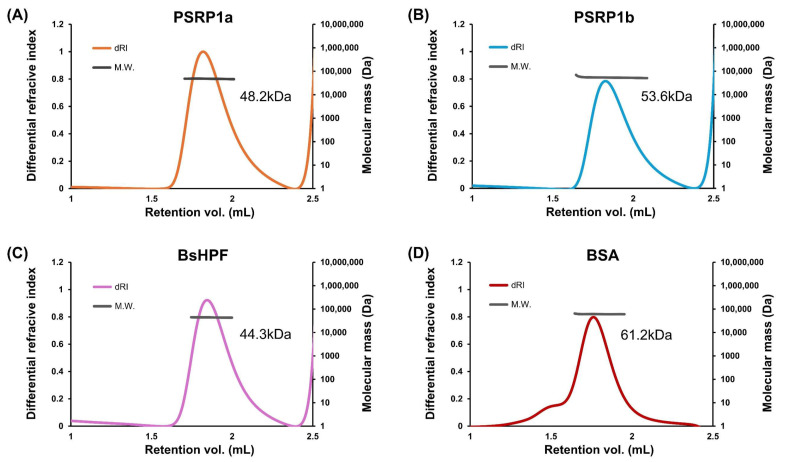
Analysis of oligomeric states of PSRP1a, PSRP1b, and BsHPF. (**A**–**D**) PSRP1a (**A**), PSRP1b (**B**), BsHPF, a dimeric control protein (**C**), and BSA, a monomeric control protein (**D**), had molecular weights of 48.2, 53.6, 44.3, and 61.2 kDa, respectively. The differential refractive index (dRI) trace indicates the protein elution profile, and the molecular weight (MW) at each elution point was determined using multi-angle light scattering (MALS).

**Figure 5 plants-14-03155-f005:**
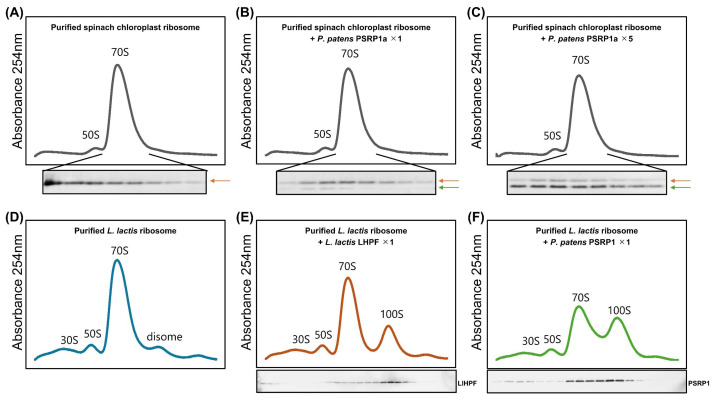
In vitro analysis of 100S dimer formation in spinach chloroplast and *Lactococcus lactis* ribosomes with the addition of PSRP1a or LlHPF. (**A**–**C**) Sucrose density profile of purified spinach chloroplast ribosomes, with PSRP1a added at 1:1 (**B**) and 1:5 (**C**) ratios to ribosomes. The Western blot show PSRP1; the orange arrow indicates endogenous spinach PSRP1 while the green arrow indicates added purified *P. patens* PSRP1a. (**D**–**F**) Sucrose density profile of purified *L. lactis* ribosomes alone (**D**), with *L. lactis* LHPF (LlHPF) added (**E**), or with PSRP1a added (**F**), each at a 1:1 ratio. The Western blot show LlHPF (**E**) and PSRP1a (**F**).

**Table 1 plants-14-03155-t001:** Comparison of calculated and experimentally determined molecular weights of PSRP1a, PSRP1b, BsHPF, and BSA by SEC-MALS.

Protein	Mass (kDa)		State in Solution
	Calculated	SEC-MALS	
PSRP1a	26.8	48.2	Dimer
PSRP1b	26.8	53.6	Dimer
BsHPF	24.2	44.3	Dimer
BSA	66.4	61.2	Monomer

## Data Availability

The original contributions presented in this study are included in the article/Appendix A, and further inquiries can be directed to the corresponding author.

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
