# Peer review of "Chloroplast Hibernation-Promoting Factor PSRP1 Prevents Ribosome Degradation Under Darkness Independently of 100S Dimer Formation"

_plants, 2025, doi:10.3390/plants14203155_

Round 1

Reviewer 1 Report

Comments and Suggestions for Authors

The manuscript examines PSRP1 in Physcomitrium patens and shows that PSRP1 levels increase in darkness, where the protein protects chlororibosomes from rRNA degradation. PSRP1 knockouts grow more slowly and despite PSRP1 being a dimer in solution, 100S dimers are not observed, however PSRP1 can induce 100S with bacterial ribosomes in vitro. Based on prior cryo-EM work of chlororibosome and PSRP1, the authors argue that preservation is distinct from bacterial 100S-mediated hibernation.

Comments:

1. Introduction: the central mechanistic insight that bS1c sterically blocks the dimer interface, so chlororibosomes do not form 100S, has already been established by the spinach cryo-EM structure. Please set this explicitly in the final paragraph, so readers understand what is known structurally and what this paper adds functionally.
Add: “Cryo-EM of spinach chlororibosomes with PSRP1 revealed that bS1c is reduced and tightly bound via an N-terminal extension, precluding HPF-like chlororibosome dimerization by occupying the interface (10.1038/s41477-018-0129-6).”

2. Results: line 207 , add a one-sentence bridge to the prior structure: “This is consistent with the spinach chlororibosome structure, in which bS1c occupies the anticipated dimer interface, thereby blocking 100S formation (10.1038/s41477-018-0129-6).”

3. Results: lines 209-211, replace with: “Thus, while PSRP1 retains intrinsic 100S-promoting capacity on bacterial ribosomes, chloroplastic bS1c likely prevents dimerization, in agreement with the prior structural model (10.1038/s41477-018-0129-6).”

4.  Figures: because the structure is from spinach while experiments are in P. patens, please add a multiple sequence alignment showing that the bS1c N-terminal extension implicated in tight ribosome binding is conserved.

• Do you consider the topic original or relevant to the field? Does it address a specific gap in the field? Please also explain why this is/ is not the case.
It addresses a gap because PSRP1 hasn’t been studied from that angle.

• What does it add to the subject area compared with other published material?
Other published work showed using cryo-EM that PSRP1 binds to the chlororibosome via bS1c, which occupies the dimer interface and thereby blocks 100S formation. The current manuscript connects those structural data to function.

• What specific improvements should the authors consider regarding the methodology?
I listed three edits and suggested one additional figure to expand the conclusions beyond a single organism.

• Are the conclusions consistent with the evidence and arguments presented and do they address the main question posed? Please also explain why this is/is not the case.
Yes.

• Are the references appropriate?
Generally ok, and I suggested citing previous work in three places where those results are mentioned but not cited: the last paragraph of the Introduction, line 207, and line 211.

• Any additional comments on the tables and figures.
As mentioned, a figure with a multiple sequence alignment would be helpful.

Author Response

Response to reviewers’ comments

We thank the reviewers for spending precious time reviewing our manuscript and providing valuable insights. Please find detailed responses below and the corresponding revisions/corrections highlighted/in track changes in the re-submitted files.

Reviewer #1

Comments 1: Introduction: the central mechanistic insight that bS1c sterically blocks the dimer interface, so chlororibosomes do not form 100S, has already been established by the spinach cryo-EM structure. Please set this explicitly in the final paragraph, so readers understand what is known structurally and what this paper adds functionally.

Add: “Cryo-EM of spinach chlororibosomes with PSRP1 revealed that bS1c is reduced and tightly bound via an N-terminal extension, precluding HPF-like chlororibosome dimerization by occupying the interface (10.1038/s41477-018-0129-6).”

Response 1: We appreciate the reviewer’s suggestion. As recommended, we revised the final paragraph of the Introduction to explicitly state that the spinach cryo-EM structure has demonstrated that bS1c sterically blocks the dimer interface, thereby preventing 100S formation (Lines 77-79). We also added the suggested sentence with the appropriate reference (10.1038/s41477-018-0129-6) in the revised manuscript (Lines 77-79). This addition clarifies the structural background and emphasizes that our study specifically addresses the functional role of PSRP1.

Comments 2: Results: line 207, add a one-sentence bridge to the prior structure: “This is consistent with the spinach chlororibosome structure, in which bS1c occupies the anticipated dimer interface, thereby blocking 100S formation (10.1038/s41477-018-0129-6).”

Response 2: We thank the reviewer for this helpful suggestion. As recommended, we added a bridging sentence to the Results section (Lines 232-234) to link our functional findings with the established structural background, together with the appropriate reference (10.1038/s41477-018-0129-6).

Comments 3: Results: lines 209-211, replace with: “Thus, while PSRP1 retains intrinsic 100S-promoting capacity on bacterial ribosomes, chloroplastic bS1c likely prevents dimerization, in agreement with the prior structural model (10.1038/s41477-018-0129-6).”

Response 3: We thank the reviewer for this valuable suggestion. As recommended, we replaced the text in the Results section (Lines 236-238) with the sentence provided by the reviewer, together with the appropriate reference (10.1038/s41477-018-0129-6).

Comments 4: Figures: because the structure is from spinach while experiments are in P. patens, please add a multiple sequence alignment showing that the bS1c N-terminal extension implicated in tight ribosome binding is conserved.

Response 4: We thank the reviewer for this constructive suggestion. As recommended, we added a multiple sequence alignment of bS1c from Physcomitrium patens, spinach, and Arabidopsis in the supplementary materials (Figure S6). In addition, to complement the sequence analysis and examine structural conservation, we performed a structural comparison using Alphafold3. By comparing the predicted structures of P. patens and spinach bS1c, we confirmed that the N-terminal extension is structurally conserved, thereby supporting the functional relevance of our experiments. We also added a description of these analyses to the Discussion (Lines 288-292).

Reviewer 2 Report

Comments and Suggestions for Authors

This is an interesting and well-presented study investigating the function of chloroplast ribosome hibernation-promoting factors (PSRP1) in Physcomitrium. The authors build on earlier work in higher plants, where psrp1 mutants lacked discernible phenotypes, by generating double knockouts of two paralogs in Physcomitrium. This strategy reveals a clear growth phenotype, making a strong case that these factors have physiological importance in at least some plant lineages.

The authors show convincingly that PSRP1 accumulates in darkness and that its loss leads to reduced accumulation of 16S and 23S rRNA. This represents the first clear molecular phenotype associated with PSRP1 in plants. Additionally, the heterologous experiment demonstrating that PSRP1 can promote 100S ribosome dimer formation of purified E. coli ribsoomes, but not with purified chloroplast ribosomes, is intriguing and raises important questions about the divergence of ribosome regulatory mechanisms between bacteria and chloroplasts. A further demonstration that PSRP1 forms dimers in vitro adds to to the biochemical characterization of this fact, although a similar behaviour has been demonstrated by related proteins in the past.

Overall, this is a valuable contribution that highlights functional differences in PSRP functions across taxa and provides an important entry point for further mechanistic studies. The work is generally well supported by the data, though in some places the conclusions should be stated more cautiously, particularly with respect to the mechanistic basis of rRNA reduction.

Major point

The most intriguing molecular observation is the reduction in rRNA levels in the psrp1 double knockout. The authors interpret this as evidence of rRNA degradation. However, conceptually, the reduction could also arise from decreased transcription or defects in rRNA processing, given that chloroplast RNAs undergo extensive processing by multiple nucleases and RNA-binding proteins.

The current analysis relies on qPCR, which does not provide information about RNA processing, integrity, or degradation intermediates. To distinguish between degradation and reduced synthesis, additional analysis is required. A straightforward experiment would be to perform RNA gel blot hybridization with probes against 16S and 23S rRNAs. This approach would not only confirm the reduction in steady-state rRNA levels but also reveal potential processing defects or degradation intermediates, thereby clarifying the molecular basis of the observed phenotype.

Minor points

Abstract, line 22: The statement “mutant displayed extensive chloroplast RNA degradation” is not formally demonstrated. Please rephrase to “reduction in rRNA content.”

Abstract, line 26: The claim that the study provides insights into stress tolerance mechanisms is overstated. No mechanistic insights into stress tolerance are provided, and prolonged darkness is not a physiologically relevant stress. In addition, the word “significant” should be reserved for cases where statistical significance is demonstrated. Please rephrase.

Line 234: The statement that “PSRP1 protects ribosomes from degradation” is too strong. The observed phenotype could also result from defects in ribosome biogenesis or impaired rRNA expression. This claim should be toned down.

Author Response

Response to reviewers’ comments

We thank the reviewers for spending precious time reviewing our manuscript and providing valuable insights. Please find detailed responses below and the corresponding revisions/corrections highlighted/in track changes in the re-submitted files.

Reviewer #2

Comments 1: The most intriguing molecular observation is the reduction in rRNA levels in the psrp1 double knockout. The authors interpret this as evidence of rRNA degradation. However, conceptually, the reduction could also arise from decreased transcription or defects in rRNA processing, given that chloroplast RNAs undergo extensive processing by multiple nucleases and RNA-binding proteins.

The current analysis relies on qPCR, which does not provide information about RNA processing, integrity, or degradation intermediates. To distinguish between degradation and reduced synthesis, additional analysis is required. A straightforward experiment would be to perform RNA gel blot hybridization with probes against 16S and 23S rRNAs. This approach would not only confirm the reduction in steady-state rRNA levels but also reveal potential processing defects or degradation intermediates, thereby clarifying the molecular basis of the observed phenotype.

Response 1: We agree with the reviewer that the observed reduction in rRNA levels could, in principle, result from decreased transcription or defects in rRNA processing, in addition to degradation. To address this concern, we had previously performed RNA gel blot analyses using probes against 16S and 23S rRNAs. These experiments, which we have now included as Fig. S3A in Supplementary Materials, confirmed an overall reduction of rRNA levels in the PSRP1a/b double knockout, consistent with the RT-qPCR results presented in the manuscript. We did not detect clear processing defects or stable intermediate fragments. However, since the dark treatment lasted for more than one day, we consider it possible that transient intermediates, if generated, were subsequently degraded and therefore escaped detection.

We also added to the Supplementary Materials (Figure S3B) data showing that chloroplast ribosomal protein S1 (bS1c) levels are also reduced in the PSRP1a/b strains under dark conditions. This additional observation argues against a scenario in which transcriptional repression or processing defects alone explain the observed reduction of rRNA and ribosomal protein levels. Instead, it supports the idea that PSRP1 functions as a chloroplast ribosome hibernation factor that helps stabilize chloroplast ribosomes and prevent their degradation in the dark.

While we acknowledge that alternative explanations such as reduced transcription or impaired processing cannot be fully excluded, we have explicitly addressed these possibilities, along with other limitations of our study, in the revised Discussion (Lines 295–302).

Comments 2: Abstract, line 22: The statement “mutant displayed extensive chloroplast RNA degradation” is not formally demonstrated. Please rephrase to “reduction in rRNA content.”

Response 2: We appreciate the reviewer’s comment. As suggested, we have revised the wording in the Abstract (lines 22–23) from “extensive chloroplast rRNA degradation” to “reduction in rRNA content”.

Comments 3: Abstract, line 26: The claim that the study provides insights into stress tolerance mechanisms is overstated. No mechanistic insights into stress tolerance are provided, and prolonged darkness is not a physiologically relevant stress. In addition, the word “significant” should be reserved for cases where statistical significance is demonstrated. Please rephrase.

Response 3: We appreciate the reviewer’s helpful comment. As suggested, we have revised the wording in the Abstract (lines 26-27) to avoid overstated claims. Specifically, we removed the term “significant” and rephrased the sentence from “could provide significant insights into plant stress tolerance mechanisms” to “could offer new insights into plant stress tolerance.”

Comments 4: Line 234: The statement that “PSRP1 protects ribosomes from degradation” is too strong. The observed phenotype could also result from defects in ribosome biogenesis or impaired rRNA expression. This claim should be toned down.

Response 4: We appreciate the reviewer’s insightful comment. As suggested, we have revised the text at line 266 to more cautiously describe the function of PSRP1, which now reads as follows: “PSRP1 … contributed to maintaining chloroplast ribosome content.”

Reviewer 3 Report

Comments and Suggestions for Authors

In this work, the authors generated the PSRP1a/b double-knockout mutant in the model organism Physcomitrium patens. The deletion of PSRP1a/b leads to mild growth defects, while the growth retardation was more pronounced for PSRP1a/b mutant when co-cultured with wild-type control. In wild-type plants, PSRP1 protein accumulation increases upon prolonged dark growth condition. Interestingly, PSRP1 proteins could form dimers and induce 100S dimer formation of bacterial ribosomes in vitro. Although the genetic analyses of PSRP1 have been conducted in cyanobacteria and maize, this work provides additional insights into the importance of PSRP1 in moss plant growth and its potential involvement in ribosome stabilization during dark growth conditions.  

Major concerns:

  1. Previous structure studies have shown that PSRP1 is closely related to LrtA protein in cyanobacteria. The authors should provide sufficient background information regarding the previous genetic studies of PSRP1 in cyanobacteria and maize in the introduction. In addition, it would be beneficial if the authors explain the advantages and interesting perspective of studying PSRP1 in moss compared to other organisms.
  2. The authors detected an increased protein accumulation level of PSRP1 upon prolonged dark growth condition (Fig. 2B). To demonstrate this protein overaccumulation is specific to PSRP1, chloroplast ribosomal proteins (from both large and small subunits) and core subunits of major photosynthetic complexes should be included in the western analyses as well.
  3. The authors suggested that PSRP1 prevents chloroplast rRNA degradation in the dark based on the RT-qPCR analyses and quantification of chloroplast 16S and 23S rRNAs (Fig. 2C-D). Since both 16S and 23S rRNAs are highly abundance and undergo massive processing and maturation, the findings would be more robust if the authors could provide RNA gel blot analyses of 16S and 23S rRNAs to confirm the RT-qPCR results. Additionally, if authors propose PSRP1 is involved in stabilizing chloroplast ribosomes upon dark exposure, western analyses of chloroplast ribosomal proteins for PSRP1a/b mutant and WT control in dark condition are required.

Minor issues:

  1. The method section for western blotting is missing a description of western signal quantification process. The authors should provide details regarding the software used for this analysis.
  2. I suggest that the authors change Figure S2 to a main figure.

Author Response

Response to reviewers’ comments

We thank the reviewers for spending precious time reviewing our manuscript and providing valuable insights. Please find detailed responses below and the corresponding revisions/corrections highlighted/in track changes in the re-submitted files.

Reviewer #3

Comments 1: Previous structure studies have shown that PSRP1 is closely related to LrtA protein in cyanobacteria. The authors should provide sufficient background information regarding the previous genetic studies of PSRP1 in cyanobacteria and maize in the introduction. In addition, it would be beneficial if the authors explain the advantages and interesting perspective of studying PSRP1 in moss compared to other organisms.

Response 1: We thank the reviewer for this helpful suggestion. In the revised Introduction, we have now added background information on the cyanobacterial LrtA protein (lines 62-68), as well as an explanation of the evolutionary perspective and advantages of studying PSRP1 in P. patens (lines 85-87). Regarding the maize studies, we did not add these to the Introduction because their findings partly contradict our results, and we had already addressed these discrepancies in the Discussion. For this reason, we chose not to duplicate the description in the Introduction.

Comments 2: The authors detected an increased protein accumulation level of PSRP1 upon prolonged dark growth condition (Fig. 2B). To demonstrate this protein overaccumulation is specific to PSRP1, chloroplast ribosomal proteins (from both large and small subunits) and core subunits of major photosynthetic complexes should be included in the western analyses as well.

Response 2: We appreciate the reviewer’s suggestion to compare PSRP1 accumulation with additional ribosomal proteins and photosynthetic subunits. In response, we have newly included supplementary data (Figure S3B) showing immunoblot analysis of chloroplast ribosomal protein S1 (bS1c). Unlike PSRP1, bS1c did not exhibit a marked increase in wild-type plants under prolonged darkness, indicating that the observed overaccumulation is not a general feature of chloroplast ribosomal proteins. We agree that extending the analysis to a broader set of ribosomal and photosynthetic proteins will be valuable. However, we currently do not have the necessary antibodies for these targets. We consider this an important direction for future research.

Comments 3: The authors suggested that PSRP1 prevents chloroplast rRNA degradation in the dark based on the RT-qPCR analyses and quantification of chloroplast 16S and 23S rRNAs (Fig. 2C-D). Since both 16S and 23S rRNAs are highly abundance and undergo massive processing and maturation, the findings would be more robust if the authors could provide RNA gel blot analyses of 16S and 23S rRNAs to confirm the RT-qPCR results. Additionally, if authors propose PSRP1 is involved in stabilizing chloroplast ribosomes upon dark exposure, western analyses of chloroplast ribosomal proteins for PSRP1a/b mutant and WT control in dark condition are required.

Response 3: We appreciate the reviewer’s valuable suggestion. As requested, we added to the Supplementary Materials of RNA gel blot analyses using probes against chloroplast 16S and 23S rRNAs (Figure S3A). These results confirmed a reduction of both Cp16S and Cp23S rRNAs in the psrp1a/b mutant under dark conditions, consistent with the RT-qPCR results presented in the manuscript.

In addition, as described in Response 2, we have newly included supplementary data (Figure S3B) showing immunoblot analysis of chloroplast ribosomal protein S1 (bS1c). Similar to rRNA, bS1c levels also exhibited a reduction in the PSRP1a/b strains under prolonged darkness, supporting the idea that PSRP1 contributes to maintaining chloroplast ribosome integrity.

Comments 3: The method section for western blotting is missing a description of western signal quantification process. The authors should provide details regarding the software used for this analysis.

Response 3: We thank the reviewer for pointing this out. In response, we have revised Section 4.3 of the Methods to include a description of the western blot signal quantification procedure (lines 373-374).

Comments 4: I suggest that the authors change Figure S2 to a main figure.

Response 4: We appreciate the reviewer’s suggestion. As recommended, we have moved the original Supplementary Figure S2 to the main figures and designated it as Figure 1 in the revised manuscript.

Round 2

Reviewer 2 Report

Comments and Suggestions for Authors

All my concerns were addressed. 

Author Response

We thank the reviewer for the constructive comments provided throughout the review process

Reviewer 3 Report

Comments and Suggestions for Authors

The authors have addressed the major points raised in the last review.

Minor issues:

  1.      I found the newly included RNA gel analysis of chloroplast rRNAs and bS1c western results in Figure S3 are very interesting and supportive to the findings in the manuscript. Therefore, I suggest adding these results as panels E and F in the main figure 3.  
  2.       For RNA gel blot analysis in of chloroplast rRNAs, methylene blue staining of membranes is necessary to verify the equal loading of different total RNA samples. 
  3. The method section for RNA gel blot analysis is missing.

Author Response

Response to reviewers’ comments

We thank the reviewers for spending precious time reviewing our manuscript and providing valuable insights. Please find detailed responses below and the corresponding revisions/corrections highlighted/in track changes in the re-submitted files.

Reviewer #3

Comments 1: I found the newly included RNA gel analysis of chloroplast rRNAs and bS1c western results in Figure S3 are very interesting and supportive to the findings in the manuscript. Therefore, I suggest adding these results as panels E and F in the main figure 3.

Response 1: We appreciate the reviewer’s positive comment and valuable suggestion. In accordance with the reviewer’s advice, we have moved the RNA gel blot and bS1c Western blot results (previously shown in Figure S3) into the main Figure 3 as new panels E and F.

Comments 2: For RNA gel blot analysis in of chloroplast rRNAs, methylene blue staining of membranes is necessary to verify the equal loading of different total RNA samples. 

Response 2: We appreciate the reviewer’s suggestion. Although methylene blue staining of membranes was not performed, we verified equal RNA loading by showing the ethidium bromide–stained gel image before blotting. The cytosolic 28S rRNA band was used as a loading control, as indicated in the revised figure legend.

Comments 3: The method section for RNA gel blot analysis is missing.

Response 3: We thank the reviewer for pointing this out. We have now added a detailed Methods subsection titled “4.6. Northern blotting” describing the RNA gel blot procedure.